# Myo-Inositol Oxygenase (MIOX): A Pivotal Regulator and Therapeutic Target in Multiple Diseases

**DOI:** 10.3390/cimb47090745

**Published:** 2025-09-11

**Authors:** Shaocong Han, Min Zhang, Huan Yang, Huiqiong Yang, Yanmei Tang, Weixi Li, Li Li, Jie Yu, Xingxin Yang

**Affiliations:** 1The First Clinical College, Yunnan University of Chinese Medicine, 1076 Yuhua Road, Kunming 650500, China; 2College of Pharmaceutical Science, Yunnan University of Chinese Medicine, 1076 Yuhua Road, Kunming 650500, China; 3School of Basic Medical Sciences, Yunnan University of Chinese Medicine, 1076 Yuhua Road, Kunming 650500, China

**Keywords:** MIOX, myo-inositol catabolism, dysregulation, pathological consequences

## Abstract

Myo-inositol oxygenase (MIOX), as the sole enzyme catalyzing myo-inositol (MI) catabolism in mammals, plays a central role in maintaining intracellular MI homeostasis. Dysregulation of MIOX activity disrupts MI metabolic balance, leading to pathological processes including oxidative stress, inflammation, and ferroptosis, which subsequently induce multiple diseases such as metabolic syndrome, neurological disorders, tumors, and reproductive/developmental disorders. This article systematically reviews the structure and function of MIOX as well as the pathological consequences arising from its dysregulation. Although its pathological significance is increasingly recognized, the molecular mechanisms of MIOX in many diseases have not been fully elucidated, and targeted modulators of MIOX are lacking. Future research should focus on the in-depth elucidation of the pathogenic mechanisms of MIOX disorders and the development of MIOX modulators, thereby providing precise therapeutic strategies for related diseases.

## 1. Introduction

Myo-inositol oxygenase (MIOX) is recognized as the key enzyme responsible for myo-inositol (MI) metabolism in mammals, with its activity being critically important for the maintenance of intracellular MI homeostasis. MI is a cyclitol naturally present in mammalian tissues, higher plants, fungi, and certain bacterial species [1]. It is present inside cells in the free form and in the plasma membrane as phosphatidylinositols, which are phosphorylated into phosphatidylinositol phosphate and phosphatidylinositol biphosphate (PIP2) through a cascade of phosphorylation reactions. Cleavage of PIP2 by phospholipase-C leads to inositol triphosphate (IP3), a key second messenger [2]. As a pivotal mediator in eukaryotic signal transduction, IP3 is responsible for the regulation of Ca^2+^ release, thereby orchestrating fundamental physiological processes such as cellular proliferation, apoptosis, and energy metabolism [3,4]. Despite extensive research on inositol phosphates and phosphoinositides, the dynamic regulation mechanisms governing total cellular MI levels and the pathological significance of their homeostatic imbalance have yet to be fully elucidated.

As the sole enzyme responsible for MI catabolism, MIOX mediates the catabolism of MI through the glucuronate–xylulose pathway [5,6]. MI is oxidized to D-glucuronic acid (DG) via MIOX, which is subsequently converted into D-xylulose 5-phosphate via a multistep enzymatic cascade, ultimately feeding into the pentose phosphate pathway (PPP), as shown in Figure 1. PPP is recognized not only as a provider of biosynthetic precursors but also as the core system for maintaining cellular redox homeostasis. This establishes it as a critical metabolic hub [1]. However, dysregulated MIOX activity disrupts the homeostasis of the MI metabolic network, triggering key physiological processes such as oxidative stress [7], inflammatory responses [8], ferroptosis [9], and epigenetic regulation [10]. This leads to the induction of multi-system diseases such as metabolic syndrome, neurological diseases, tumors, and reproductive and developmental disorders. To summarize, disruption of MIOX activity will lead to serious pathological consequences. Targeted regulation of MIOX is anticipated to emerge as a novel therapeutic strategy to delay the progression of these diseases.

Given the significance of MIOX, this review aims to systematically summarize the structure and function of MIOX as well as the pathological consequences and molecular mechanisms of its dysregulation across various pathological conditions. This will provide theoretical support and potential research directions for future studies on the treatment of related diseases targeting MIOX.

## 2. Characterization and Biological Function of MIOX

### 2.1. Characterization of MIOX

MIOX, a 33 kDa cytosolic enzyme, is highly conserved among mammals, with 91% sequence homology between humans and mice [11]. It is predominantly expressed in the kidney, liver, and neural tissues, where it is recognized as the exclusive catalyst for MI catabolism [1,12,13]. MIOX utilizes a mixed-valence non-heme diiron center Fe(II)/Fe(III) to activate its substrates, MI and O_2_. Evidence suggests that the Fe(III) site coordinates MI via its C1 and C6 hydroxyl groups, the Fe(II) site reversibly coordinates O_2_ to produce a superoxo-diiron(III/III) intermediate, and the pendant oxygen atom of the superoxide ligand abstracts hydrogen from C1 to initiate the unique C-C-bond-cleaving, four-electron oxidation reaction. This reaction cascade is ultimately responsible for the conversion of MI to DG, and the mixed-valence state is regenerated after each cycle of catalysis [14,15,16].

The MIOX gene exhibits high conservation across primates, indicating its critical physiological role throughout evolution and suggesting that sequence divergence is predominantly governed by purifying selection to prevent the accumulation of deleterious mutations [17].

The enzymatic activity of MIOX is dependent on MI and Fe^2+^ ions. Its catalytic efficiency is directly modulated by its oligomerization state, particularly the transition from monomer to multimer. Furthermore, substrate (MI) binding induces conformational changes that promote oligomerization. This enhanced oligomerization facilitates the enzyme’s binding capacity for the cofactor (Fe^2+^), thereby regulating its own oxidative efficiency [18]. Dysregulation of MIOX function is directly implicated in perturbations of the inositol pool. MI is recognized not only as a precursor for cellular signaling molecules (e.g., IP3) but also as a critical mediator of osmoregulation, lipid metabolism, and mitochondrial fission [19,20,21]. Through its regulation of MI degradation, MIOX plays a central role in maintaining cellular inositol homeostasis and exerts a profound influence on aforementioned cellular functions such as signal transduction, osmoregulation, and energy metabolism. Furthermore, MIOX expression and activity are modulated by organic osmolytes, high glucose, fatty acids, and oxidant stress. Interestingly, phosphorylation of serine/threonine residues in MIOX has been shown to augment enzymatic activity via post-translational modification [22,23]. Therefore, dysfunction of MIOX directly disrupts the inositol pool and consequently perturbs these interconnected physiological processes, establishing a molecular link to various diseases and providing a theoretical basis for research on MIOX-targeted interventions.

### 2.2. The Function of MIOX

MIOX is a pivotal metabolic enzyme that catalyzes the conversion of MI and O_2_ to DG. Beyond its canonical role in carbohydrate metabolism, MIOX functions as a central player in diverse pathophysiological processes. By modulating redox homeostasis, endoplasmic reticulum stress, ferroptosis, and specific cellular signaling pathways (Figure 2), MIOX is critically involved in disease progression.

#### 2.2.1. Redox Equilibrium

The catalytic process mediated by MIOX is driven through its conserved Fe(II)/Fe(III) active center, converting MI and O_2_ into DG. This biochemical reaction holds a pivotal position in MI metabolism and the glucuronate–xylulose pathway, and its indispensable role in sustaining complex metabolic networks in vivo has been extensively recognized. During this process, interconversion of coenzymes NADPH/NADP+ and NADH/NAD+ is concomitantly observed. The diiron center of MIOX is maintained in a state of structural and functional stability through coordination by histidine and aspartate residues, thereby sustaining metabolic homeostasis in the organism [24].

Research has shown that the disruption of MIOX’s enzymatic activity directly interferes with the balance of NADH/NAD+ ratio, consequently altering cellular antioxidant capacity. This dysregulation is further shown to impair the cellular response to oxidative stress and compromise overall metabolic homeostasis [25]. Furthermore, MIOX is recognized as playing a central role in modulating cellular redox balance, with its functional scope encompassing the generation of reactive oxygen species (ROS) and regulation of antioxidant defenses. These mechanisms are particularly implicated in renal pathologies and metabolic disorders [7]. Overexpression of MIOX has been shown to significantly enhance cadmium-induced intracellular ROS accumulation in tissues [26,27] and to indirectly reduce the levels of antioxidant molecules such as glutathione (GSH) and glutathione peroxidase [28]. In diabetic nephropathy (DN) models, hyperglycemic conditions are reported to promote hypomethylation-dependent binding of the transcription factor Sp-1 to the MIOX promoter, resulting in upregulation of MIOX expression. This aberrant regulation is associated with exacerbated mitochondrial ROS production and aggravated oxidative damage in renal tubules [29]. Additionally, MIOX-mediated activation of the ROS/ALOX-12/12-HETE/GPR31 signaling pathway is identified as a mechanism driving lipid peroxidation, thereby intensifying gentamicin-induced nephrotoxicity [30]. Collectively, MIOX is established as an indispensable regulator of intracellular redox equilibrium, with its dysregulation mechanistically linked to oxidative-stress-related tissue injury and metabolic dysfunctions.

#### 2.2.2. Endoplasmic Reticulum Stress

In addition to oxidative stress, endoplasmic reticulum stress (ERS) is implicated in the pathogenesis of various metabolic disorders. ERS, recognized as a critical cellular defense mechanism against the accumulation of unfolded/misfolded proteins, is dynamically regulated through the activation of three unfolded protein response (UPR) signaling pathways: IRE1α, PERK, and ATF6 [31]. Recent investigations have revealed that MIOX exerts direct or indirect regulatory effects on UPR signaling via its modulation of redox homeostasis. Specifically, MIOX overexpression has been demonstrated to exacerbate ERS-induced renal tubulointerstitial injury by upregulating ERS-associated proteins (GRP78, sXBP1, CHOP), pro-apoptotic factors, and inflammatory cytokines, and by augmenting ROS production [32].

#### 2.2.3. Ferroptosis

Ferroptosis is defined as an iron-dependent, non-apoptotic form of cell death, characterized by the uncontrolled accumulation of lipid peroxidation and inhibition of glutathione peroxidase 4 (GPX4) activity. MIOX, through its regulation of redox homeostasis and lipid metabolism, has been identified as a critical regulator of ferroptosis and influences a variety of diseases. In the studies conducted by Wang et al., MIOX was demonstrated to be a key N6-methyladenosine-regulated ferroptosis gene in patients with type 2 diabetes mellitus MIOX overexpression has been shown to promote ROS accumulation, induce lipid peroxidation, and suppress both GPX4 activity and NADPH levels, thereby exacerbating ferroptosis in pancreatic β cells [33]. Furthermore, MIOX is implicated in aggravating cisplatin-induced renal injury through mechanisms involving the promotion of ferritinophagy, inhibition of GPX4 activity, and depletion of antioxidant molecules (GSH and NADPH), which collectively amplify lipid peroxidation and ferroptosis [9]. In papillary thyroid carcinoma (PTC) tissues, downregulation of MIOX expression has been observed, with its regulatory effects on iron metabolism and antioxidant defense systems suggested to play roles in PTC progression and ferroptosis modulation [34]. As a ferroptosis-related gene, MIOX has been incorporated into prognostic models for papillary renal cell carcinoma, where its low expression is correlated with poor clinical outcomes, mediated by affecting the pentose phosphate metabolism pathway and oxidative stress [35]. In hepatocellular carcinoma (HCC), the long non-coding RNA NEAT1 has been reported to indirectly upregulate MIOX expression by modulating miR-362-3p, thereby promoting ferroptosis in HCC cells [36]. Notably, MIOX exhibits dual regulatory properties in renal pathologies: in immunoglobulin A nephropathy models, MIOX was classified as a ferroptosis-associated differentially expressed gene, with its encoded enzyme potentially mediating redox enzymatic activity to exacerbate disease progression through ferroptosis-related pathways [37]. Conversely, in clear cell renal cell carcinoma (ccRCC), MIOX was categorized as a cuproptosis-associated ferroptosis gene, with its downregulation in high-risk subgroups proposed to function as a protective factor predictive of patient prognosis [38].

#### 2.2.4. Cellular Signaling Transduction

MIOX has been established as a multi-pathway regulatory hub, participating in diverse pathophysiological processes through intricate signaling networks. In models of infectious cardiac dysfunction, MIOX has been shown to exacerbate inflammatory cascades by promoting NLR family pyrindomain -containing 3 (NLRP3) inflammasome assembly and interleukin-1β release [8]. Under hyperglycemic conditions, MIOX expression is upregulated, which has been linked to disruption of the mitochondrial quality control axis, suppression of the PINK1/Parkin pathway, and impaired mitophagy. These alterations are associated with accumulated dysfunctional mitochondria, exacerbating oxidative stress and cellular damage [7]. Additionally, MIOX has been reported to disrupt cellular energy homeostasis by inhibiting the AMPK/sirtuins/PGC-1α/YY-1 pathway, thereby aggravating obesity-associated renal tubular injury [39]. In obesity-related MetS, MIOX expression is upregulated by fatty acids through the mTORC1/SREBP1 signaling axis, further perturbing renal tubular energy metabolism. This mechanism is proposed to provide a therapeutic rationale for targeted interventions, such as mTORC1 inhibitors or MIOX antagonists [40]. MIOX is also identified as a critical effector molecule downstream of the AGE-RAGE signaling pathway, driving oxidative tubulointerstitial injury and fibrosis in DN via the PI3K/AKT/NF-κB/TGF-β axis and ROS-positive feedback loops [41]. Notably, in ccRCC, MIOX has been characterized as a pivotal regulatory molecule that suppresses tumor progression through the autophagy-ROS-STAT3/c-Myc signaling cascade [42].

## 3. Application of MIOX in Disease Diagnosis and Prediction

The dysfunction of MIOX will lead to exacerbation of oxidative stress and endoplasmic reticulum stress, abnormalities in ferroptosis, and disruption of intracellular signaling mechanisms. These pathophysiological processes further promote the occurrence and progression of various diseases (Figure 3).

### 3.1. Renal Diseases

The kidney is recognized as a target organ for obesity, diabetes, dyslipidemia, and hypertension. Consequently, chronic kidney disease (CKD) has been established as a major contributor to global morbidity and mortality. Epidemiological studies have reported an approximate 11% prevalence of CKD in high-income nations, underscoring its status as a critical public health challenge [43]. Furthermore, the kidney is disproportionately susceptible to acute kidney injury (AKI) induced by nephrotoxic agents, including heavy metals and drugs, due to its role as a primary detoxification organ.

In recent years, mechanistic studies of renal pathologies have intensified, leading to the identification of novel biomarkers and therapeutic targets. Among these discoveries, MIOX has been identified as a pivotal enzyme with critical roles in renal disease pathogenesis. Its involvement in mediating oxidative stress, metabolic dysregulation, and cellular injury mechanisms has positioned MIOX as a promising candidate for both biomarker development and therapeutic targeting in kidney injury.

#### 3.1.1. Acute Kidney Injury

Serum MIOX level has been identified as a promising potential biomarker for diagnosing AKI, with its clinical relevance strongly correlated to patient prognosis [44]. In multiple studies, MIOX has been proposed as a candidate biomarker for early AKI detection [45,46,47,48]. Through LC-MS/MS-based analyses of blood MI levels in patients with AKI and CKD, MI concentrations were found to be significantly elevated in patient plasma compared to healthy controls. Notably, in patients with AKI, MI elevation was detected approximately 33 h prior to observable creatinine changes, a finding that is directly interpreted as evidence linking MIOX dysfunction to renal pathology [49].

In investigations by Altun et al., patients with renal stones undergoing percutaneous nephrolithotomy or flexible ureteroscopy were observed to develop AKI postoperatively, with serum MIOX level fluctuations demonstrated to be more sensitive than serum creatinine in reflecting early renal injury. This supports the utility of MIOX being prioritized as a potential early diagnostic marker for AKI [50]. Following retrograde intrarenal surgery for nephrolithiasis, the safety of the procedure has been evaluated through postoperative serum MIOX monitoring, further validating its biomarker potential [51,52]. Additionally, serum MIOX levels have been shown to predict progression from community-acquired AKI to CKD post-discharge, highlighting its prognostic value in longitudinal renal outcomes [53].

#### 3.1.2. Heavy-Metal- and Drug-Induced Kidney Injury

The kidney is recognized as a critical target organ for toxic effects induced by diverse chemical agents and is particularly susceptible to injury mediated by these substances. Specifically, cadmium and cisplatin, both of which are classified as potent nephrotoxins, have been demonstrated to induce renal tissue damage through MIOX-dependent oxidative stress pathways. In animal experimental models, cadmium exposure has been shown to upregulate MIOX expression, subsequently activating ferroptosis and apoptotic processes [26,27,28], thereby exacerbating pathological manifestations of AKI. Conversely, MIOX functional deficiency has been demonstrated to attenuate cisplatin-induced AKI by reducing ROS generation [54,55].

In aristolochic acid I-induced AKI models, MIOX deficiency was found to mitigate oxidative stress and apoptosis, thereby effectively blocking the progression of AKI [56]. Similarly, in gentamicin-induced nephrotoxicity models, MIOX expression is upregulated, and its activity has been linked to aggravated tubular injury through modulation of lipoxygenase pathways [30]. These findings have provided a mechanistic rationale for targeting MIOX inhibition as a therapeutic strategy to alleviate drug-induced nephrotoxicity.

#### 3.1.3. DN

Elevated glucose levels are demonstrated to enhance MIOX enzymatic activity, which is implicated in the exacerbation of tubulointerstitial injury and progression of DN [57]. Lu et al. found that MIOX expression is increased in DN through animal models and in vitro experiments, and it may drive tubular injury by promoting epithelial–mesenchymal transition and extracellular matrix accumulation [57]. Xie et al. found that renal-specific oxidoreductase/myo-inositol oxygenase (RSOR/MIOX) in DN could cause NADH/NAD+ imbalance, thereby activating the PKC-α and MAPK signaling pathways, promoting TGF-β secretion and fibronectin deposition, and driving tubulointerstitial fibrosis [25]. Under hyperglycemic conditions, the demethylation of the MIOX gene is promoted through its binding to the Sp-1 transcription factor, thereby enhancing its transcriptional activity. Increased expression of MIOX induces mitochondrial fragmentation, loss of membrane potential, suppression of autophagy, and accumulation of ROS. These pathological alterations collectively lead to mitochondrial quality control dysregulation, ultimately exacerbating renal tubular injury [7,29]. In addition, clinical studies found that MIOX was upregulated in renal tubules of patients with DN, which positively correlated with renal tubular injury and oxidative stress and negatively correlated with Sirt1. The early elevation of MIOX levels in serum and urine is related to multiple indicators, and MIOX may be a new biomarker for early diagnosis of DN [58]. Collectively, these findings suggest that MIOX is a critical therapeutic target in DN.

#### 3.1.4. Other Kidney Diseases

Polycystic kidney disease (PKD) is recognized as one of the most prevalent inherited progressive nephropathies in humans [59]. Lu et al. found that cyst formation in PKD was the result of dysregulation of the GNAS-PI4KB-AKT axis signaling pathways and identified MIOX as a candidate marker gene for the disease. Although the precise mechanistic role of MIOX in PKD has not been thoroughly investigated, it can be speculated that the dysregulation of MIOX expression may exacerbate disease progression through oxidative stress and metabolic disorders based on corroborative evidence from other studies [59].

Focal segmental glomerulosclerosis (FSGS), characterized morphologically by podocyte injury, is recognized as a morphological manifestation of a spectrum of renal diseases. Its prevalence has been reported to rise persistently across populations, while therapeutic options remain limited. Consequently, elucidating FSGS pathophysiology and developing targeted therapies have emerged as critical research priorities [60]. In clinical studies, urinary MI levels were found to be significantly elevated in patients with FSGS compared to patients with minimal change in disease and healthy controls, with a negative correlation observed between MI levels and estimated glomerular filtration rate (eGFR). Concurrently, tissue MIOX expression was demonstrated to inversely correlate with urinary MI levels but positively correlate with eGFR. These findings suggest that MIOX may be implicated in the modulation of MI metabolism in FSGS, thereby influencing disease progression and renal function. This association provides mechanistic insights for understanding FSGS pathogenesis and developing novel therapeutic strategies [61].

### 3.2. Metabolic Disease

MIOX is recognized as a key hub in the systemic regulation of metabolic disorders through its integration of energy metabolism and redox signaling. Under hyperglycemic or hyperlipidemic microenvironments, MIOX-mediated MI degradation is demonstrated to not only perturb local tissue metabolism but also modulate systemic energy homeostasis via circulating metabolites [23,40]. Experimental evidence has positioned MIOX as a key regulatory molecule for renal tubular injury in metabolic diseases, with its expression and activity being regulated by fatty acids, insulin, and the mTORC1/SREBP1 pathway. These regulatory mechanisms are further shown to mediate pathological progression through oxidative stress cascades [40]. In hyperlipidemic conditions, MIOX upregulation is reported to suppress the AMPK/sirtuins pathway, resulting in dysfunction of metabolic sensors (e.g., PGC-1α and YY-1). This dysregulation is associated with disrupted mitochondrial biogenesis and oxidative phosphorylation, thereby exacerbating renal tubular injury [39]. In high-fat diet-fed murine models, insulin resistance and impaired metabolic health have been observed, accompanied by enhanced MIOX-mediated MI degradation in renal tissues. These alterations are mechanistically linked to insulin resistance and ectopic lipid deposition [62,63].

In metabolic disorders, the role of MIOX is not confined to DN but is extended to other metabolism-related pathologies. In metabolic-dysfunction-associated fatty liver disease (MAFLD), dysregulation of MI metabolism has been established to correlate closely with disease progression. Notably, MI supplementation at 4 g/day for 8 weeks has been demonstrated to improve insulin resistance (IR), lipid profiles, and hepatic function in patients with MAFLD and obesity [64,65,66]. Given that MIOX is recognized as the sole catabolic enzyme for MI, its altered expression is related to inositol metabolic perturbations. Experimental studies have further confirmed that in high-fat diet-fed spotted sea bass (*Lateolabrax maculatus*), hepatic MIOX expression is significantly modulated, leading to dysregulated carbohydrate metabolism and subsequent lipid accumulation in the liver [67]. In HCC cells, MIOX levels have been mechanistically linked to ferroptosis susceptibility [36]. Clinical investigations have suggested that in polycystic ovary syndrome (PCOS) an imbalance in inositol metabolism affects the microenvironment of oocytes [68]. As the exclusive enzyme catalyzing MI degradation, the abnormal function of MIOX may be associated with IR and hyperandrogenemia in PCOS [63,69]. Furthermore, MIOX has been implicated in obesity and hypertension, with its upregulated expression observed in both spontaneous hypertensive and obesity models. This upregulation is correlated with enhanced catabolism via the glucuronate-xylulose pathway and concomitant oxidative stress, thereby aggravating disease progression [62].

Metabolic dysregulation is recognized as a critical risk factor for multiple pathologies, in which abnormal expression of MIOX has been implicated in the pathogenesis and exacerbation of metabolic disorders through diverse mechanisms. The central role of MIOX in metabolic diseases is characterized by its involvement in IR, oxidative stress, inflammatory cascades, and ferroptosis, thereby providing a molecular rationale for novel therapeutic strategies targeting MIOX. Future investigations are expected to further elucidate the precise mechanistic contributions of MIOX to metabolic pathologies, with the ultimate goal of translating these insights into innovative clinical interventions.

### 3.3. Cancer Progression

The expression level of MIOX has been closely associated with disease progression and prognosis in various cancers, demonstrating complex and diverse roles across different cancer types. In patients with lung squamous cell carcinoma, elevated MIOX expression has been correlated with improved prognosis. This phenomenon may be related to its regulation of reactive oxygen species (ROS) levels and the subsequent impact on oxidative stress status, while the specific mechanism remains to be further investigated [70]. Conversely, in colorectal cancer (CRC), MIOX, as a methylation driver gene, is significantly upregulated in tumor tissues and observed to increase with disease progression. This upregulation is hypothesized to exacerbate CRC pathogenesis through oxidative-stress-mediated mechanisms, with its overexpression strongly associated with advanced tumor stages and poor clinical outcomes [10].

Glutamine metabolism promotes tumor progression through its provision of energy, nitrogen sources, and regulation of redox homeostasis while influencing the immune microenvironment. Wu et al. [71] reported that glutamine metabolic patterns in ccRCC drove tumor heterogeneity and immune remodeling. A prognostic model was subsequently established, through which MIOX was identified as one of the core risk genes. Notably, high expression of MIOX was significantly associated with poor clinical outcomes, thereby highlighting its potential value as both a prognostic biomarker and a promising therapeutic target.

Based on transcriptomic analysis of prostate adenocarcinoma (PRAD), a 12-gene metabolic prognostic model was constructed. It was revealed that MIOX exhibited significant overexpression in PRAD, which was found to be associated with adverse clinical outcomes and resistance to androgen receptor signaling inhibitor therapy. Furthermore, it was demonstrated to promote tumor progression through regulation of glucose metabolism, activation of DNA repair pathways, and remodeling of the immune microenvironment [72].

### 3.4. Nervous System Disease

The pathogenic mechanism of Cryptococcus neoformans involves the upregulation of inositol catabolism genes (MIO1/2/3 and MIOX2), which activate the MIOX pathway to efficiently degrade host-derived inositol. This process drives capsule biosynthesis and structural remodeling, thereby enhancing blood–brain barrier penetration, immune evasion capacity, and energy metabolism efficiency. The “inositol-MIOX-capsule” pathogenic axis promoted fungal colonization in the central nervous system and accelerated the progression of lethal meningitis in immunocompromised populations [73,74].

Furthermore, in bifenthrin-induced neurotoxicity models, elevated MIOX expression has been implicated in exacerbated neurological dysfunction through its interference with fatty acid signaling, neurotransmitter homeostasis, and oxidative stress responses [75]. Based on these findings, therapeutic targeting of the MIOX pathway is suggested as a novel strategy to mitigate neurotoxic outcomes.

### 3.5. Reproduction and Development

Prenatal ethanol exposure has been demonstrated to downregulate MIOX expression, resulting in impaired embryonic development. This is manifested by reduced blastocyst formation rates, compromised mitochondrial membrane potential, and upregulation of oxidative-stress-associated genes (CLU and APP). Collectively, these findings indicate that MIOX suppression is identified as a critical molecular event in ethanol-induced embryotoxicity [76]. In patients with gestational diabetes mellitus, MI metabolic disorders were shown to be potentially exacerbated by aberrant elevation of MIOX activity, resulting in MI depletion and attenuation of insulin signaling, thereby promoting insulin resistance development [77]. Studies in model organisms have further revealed that the elimination of MI catabolism in Drosophila melanogaster was associated with severe developmental defects. Dietary MI supplementation was shown to ameliorate metabolic phenotypes, including reduced adiposity and improved glycemic control, underscoring the essential role of MI homeostasis in developmental regulation [78].

### 3.6. Other

In the context of postpartum metabolic adaptation of dairy cows, differential expression of MIOX was observed to correlate with metabolic phenotypes: downregulation of MIOX in normal-metabolic-type cows was associated with alleviated metabolic burden, whereas MIOX overexpression in high-metabolic-type cows was suggested to exacerbate oxidative stress and increase hepatic injury risk through NADPH depletion [79]. Similarly, in high-yielding laying hens (LB strain) during the egg-laying period, elevated renal MIOX expression was accompanied by accumulation of oxidative stress markers, suggesting that MIOX may potentiate pathological progression under conditions of elevated metabolic demand [80]. In lipodystrophic gerbil model, an MI-deficient diet was observed to decrease MI content in intestinal and other tissues, accompanied by abnormal fat accumulation. However, the activity of MIOX was not significantly altered. These findings indicate that the role of MIOX in the disease may involve a complex metabolic network and suggest that differences in MI metabolic regulation exist between species [81].

Multiorgan dysfunction induced by microplastic exposure, particularly tire wear particles, is mediated through oxidative stress and metabolic disruption, with hepatic MIOX expression significantly upregulated in zebrafish, leading to dysregulated carbohydrate metabolism and redox imbalance [82]. In diabetic complications, hyperglycemia is shown to accelerate MI catabolism via MIOX upregulation, resulting in toxic metabolite accumulation (e.g., xylitol) and exacerbation of retinal and neural tissue damage, thereby positioning MIOX as a potential therapeutic target for diabetic complications [83]. In euryhaline fish such as Nile tilapia (*Oreochromis niloticus*), MIOX is implicated in enhancing hyperosmotic adaptability by modulating MI accumulation, energy metabolism, and antioxidant systems, with its expression regulated by osmostress-responsive transcription factors [84]. In poultry diet, phytase supplementation is observed to elevate plasma MI levels without altering renal MIOX expression in broiler chickens, suggesting the existence of non-feedback regulatory mechanisms in MI metabolism [85].

## 4. Intervention Strategies Targeting MIOX

MIOX has been identified to play a critical role in the occurrence and development of various diseases, and the development of MIOX-targeting pharmaceutical agents is of great significance. Several drugs targeting MIOX exert distinct biological effects under specific pathological conditions. For instance, mulberry leaf extract upregulates MIOX expression and improves hepatic function, while cyclo (His-Pro) and canagliflozin suppress MIOX and improve diabetes and DN, respectively (Table 1). Furthermore, MIOX has been implicated in the pathogenesis of various disorders, including acute kidney injury, cancer, neurological diseases, as well as reproductive and developmental disorders. Despite these associations, there are currently no approved therapeutic agents specifically targeting MIOX for these conditions.

## 5. Conclusions and Future Perspectives

MIOX, as a metabolic enzyme, plays a central role in various human pathological processes. However, its molecular mechanisms remain incompletely understood across numerous pathological contexts. The pathogenesis of diseases can be further elucidated through the application of emerging approaches such as genome-wide association studies, proteomics, epigenetics, as well as single-cell sequencing, spatial metabolomics, and integrated multi-omics strategies.

Another significant challenge in current research is the severe lack of small-molecule ligands designed to modulate MIOX activity or expression. The three-dimensional structures of human MIOX (PDB ID: 2IDN [88]) (Figure 2) and mouse MIOX (PDB IDs: 2HUO [24], 3BXD [88]) provide a foundational framework for structure-based drug design. Multiple approaches, such as ligand–protein molecular docking, molecular dynamics’ simulations, pharmacophore modeling, and emerging artificial-intelligence-based methods (including deep learning, graph convolutional networks, and equivariant graph neural networks) can be used to develop MIOX-targeting ligands.

Elucidating its specific molecular mechanisms in diverse diseases, along with pursuing and developing selective molecular ligands for it, would not only advance mechanistic insights into disease pathogenesis but also enable targeted therapies for MIOX-associated disorders.

## Figures and Tables

**Figure 1 cimb-47-00745-f001:**
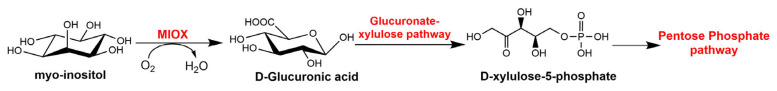
The metabolic process from myo-inositol to the pentose phosphate pathway.

**Figure 2 cimb-47-00745-f002:**
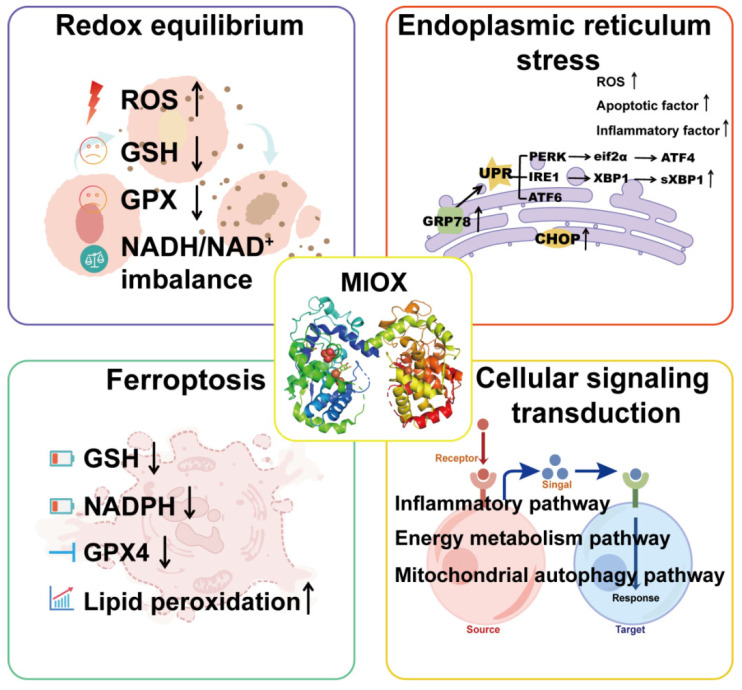
The function of MIOX.

**Figure 3 cimb-47-00745-f003:**
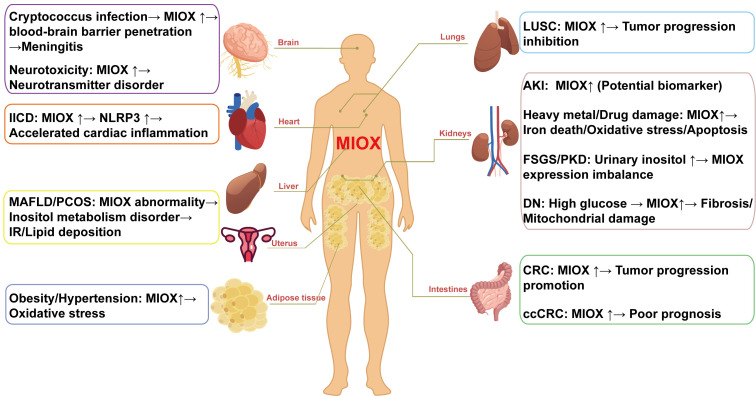
Systemicinvolvement of MIOX dysregulation in human diseases. AKI, acute kidney injury; IR, insulin resistance; IICD, infection-induced cardiac dysfunction; DN, diabetic nephropathy; FSGS, focal segmental glomerulosclerosis; PKD, polycystic kidney disease; CRC, colorectal cancer; ccRCC, clear cell renal cell carcinoma; MAFLD, metabolic-dysfunction-associated fatty liver disease; PCOS, polycystic ovary syndrome; LUSC, lung squamous cell carcinoma.

**Table 1 cimb-47-00745-t001:** Intervention strategies targeting MIOX and their biological effects.

Molecule or Formulation	Biological Effect
Mulberry leaf extract	Upregulates MIOX expression, enhances liver carbohydrate metabolism, and optimizes energy supply [67].
Cyclo (His-Pro)	Inhibits MIOX expression, improves glucose homeostasis, enhances insulin sensitivity, and indirectly alleviates inflammatory responses [86].
Canagliflozin	Inhibits MIOX expression and abnormal glycolysis, blocks glucose uptake, reduces mitochondrial ROS production, and suppresses fibrosis generation [87].

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
