# Peer review of "Myo-Inositol Oxygenase (MIOX): A Pivotal Regulator and Therapeutic Target in Multiple Diseases"

_cimb, 2025, doi:10.3390/cimb47090745_

Round 1

Reviewer 1 Report

Comments and Suggestions for Authors

The manuscript is a revision of the biological roll of MIOX, from cellular and physiological function, disease relation and diagnosis, and finally intervention strategies with different molecules and drugs, Information is well presented and resumed in three figures, however for a better understanding;

-I suggest a table for the last section, 4 Intervention strategies targeting MIOX, a table with two columns, signaling in one the molecule or formulation and in the second the biological effect.

Author Response

We are grateful for your positive comments and constructive feedback. The manuscript has been exactly revised according to your suggestions point by point. All the modifications have been highlighted in red in the revised manuscript.

Comments 1: The manuscript is a revision of the biological roll of MIOX, from cellular and physiological function, disease relation and diagnosis, and finally intervention strategies with different molecules and drugs, Information is well presented and resumed in three figures, however for a better understanding;

-I suggest a table for the last section, 4 Intervention strategies targeting MIOX, a table with two columns, signaling in one the molecule or formulation and in the second the biological effect.

Response: Thank you for this valuable suggestion. We agree with your comment. We have added a table on page 11-12, section 4, line 441, which lists the various molecules/formulations alongside their corresponding biological effects. The details are provided in Table 1 below.

In addition, to avoid redundant descriptions in the text, we have thoroughly rewritten the content of Section 4. The specific revisions can be found on page 11-12, section 4, lines 431-440 of the revised manuscript.

Table 1. Intervention strategies targeting MIOX and their biological effects

Molecule or formulation

Biological effect

Mulberry leaf extract

Upregulates MIOX expression, enhances liver carbohydrate metabolism, and optimizes energy supply [67].

Cyclo (His-Pro)

Inhibits MIOX expression, improves glucose homeostasis, enhances insulin sensitivity and indirectly alleviates inflammatory responses [86].

Canagliflozin

Inhibits MIOX expression and abnormal glycolysis, blocks glucose uptake, reduces mitochondrial ROS production, and suppresses fibrosis generation [87].

Reviewer 2 Report

Comments and Suggestions for Authors

Report on manuscript cimb-3834664 entitled “Myo-inositol oxygenase (MIOX): a pivotal regulator and therapeutic target in multiple disease” by Han et al.

The manuscript (MS) submitted for publication is a review dealing with the MIOX enzyme mostly in the context of ailments or conditions in which MIOX is (or could be) involved. The aim is described in the final sentence of the abstract, “providing precise therapeutic strategies for related diseases”.

Even through the MS is quite comprehensive, with the aim that is mentioned by the authors (vide supra) I consider that there is one aspect that is totally missing from this review: the manuscript needs a paragraph (not necessarily very long) mentioning the strategies that could be explored in order to develop novel compounds for disease treatment. In this context, even though the MS contains a Figure showing the 3-D structure of MIOX (Figure 2), no mention is made in the text of the relevant 3-D crystal structures (PDB ids 2IBN, [2HUO, 3BXD]) with the corresponding reference. This mention is important in the context of the development of new therapeutics by means of “3-D structure based design” using well established methodologies (design of new chemicals, molecular docking, Molecular Dynamics etc) as well as novel methods that involve A.I. (such as in the design of peptide binders). Such a paragraph should be present in order to introduce real perspectives for future work targeting the MIOX enzyme.

There are also a couple of points that I have noted:

In the title of the MS, I think it should read “multiple diseases” (multiple refers to several, i.e. plural whereas disease is singular).

In section 3.2, line 325 (and removing the parenthetical element of the sentence), “MIOX its abnormal function may be associated with IR and hyperandrogenemia” is poorly written. This should be rewritten.

Author Response

We are grateful for your positive comments and constructive feedback. The manuscript has been exactly revised according to your suggestions point by point. All the modifications have been highlighted in red in the revised manuscript.

Comments 1. The manuscript (MS) submitted for publication is a review dealing with the MIOX enzyme mostly in the context of ailments or conditions in which MIOX is (or could be) involved. The aim is described in the final sentence of the abstract, “providing precise therapeutic strategies for related diseases”.

Even through the MS is quite comprehensive, with the aim that is mentioned by the authors (vide supra) I consider that there is one aspect that is totally missing from this review: the manuscript needs a paragraph (not necessarily very long) mentioning the strategies that could be explored in order to develop novel compounds for disease treatment. In this context, even though the MS contains a Figure showing the 3-D structure of MIOX (Figure 2), no mention is made in the text of the relevant 3-D crystal structures (PDB ids 2IBN, [2HUO, 3BXD]) with the corresponding reference. This mention is important in the context of the development of new therapeutics by means of “3-D structure based design” using well established methodologies (design of new chemicals, molecular docking, Molecular Dynamics etc) as well as novel methods that involve A.I. (such as in the design of peptide binders). Such a paragraph should be present in order to introduce real perspectives for future work targeting the MIOX enzyme.

Response: Thank you for this valuable suggestion. We fully agree with this comment. We have added a new paragraph in the final section, Conclusions and future perspectives, to explore strategies such as 3-D structure based drug design and AI-driven approaches for developing novel compounds in disease treatment. The change can be found on page 12, section 5, line 444, line 447-450 and line 452-458 of the revised manuscript.

Comments 2: In the title of the MS, I think it should read “multiple diseases” (multiple refers to several, i.e. plural whereas disease is singular).

Response: Thank you for pointing this out. We agree with this comment. We have revised the term "multiple disease" to "multiple diseases". The change can be found in the title, line 3 of the revised manuscript.

Comments 3: In section 3.2, line 325 (and removing the parenthetical element of the sentence), “MIOX its abnormal function may be associated with IR and hyperandrogenemia” is poorly written. This should be rewritten.

Response: Thank you for pointing this out. We sincerely thank the reviewer for highlighting the lack of clarity in the original sentence. We have carefully revised the sentence to improve its grammatical structure and clarity. The original phrasing has been rephrased to: As the exclusive enzyme catalyzing MI degradation, the abnormal function of MIOX may be associated with IR and hyperandrogenemia in PCOS. This revision eliminates the dangling modifier and clarifies the subject-verb relationship, thereby enhancing readability and scientific precision. The modified sentence can be found on page 9, section 3.2, line 325-326 of the revised manuscript.